# Unveiling the Diversity of Bangka Island's Mangroves: A Baseline for Effective Conservation and Restoration

Suci Puspita Sari [1,2,*], Nico Koedam [3,4,5,6], Aditya Pamungkas [2], Muhammad Rizza Muftiadi [7] and Frieke Van Coillie [1]

1   Remote Sensing Spatial Analysis Lab (REMOSA), Department of Environment, Ghent University, 9000 Ghent, Belgium; frieke.vancoillie@ugent.be
2   Department of Marine Science, Universitas Bangka Belitung, Bangka 33172, Indonesia; adityapamungkas@ubb.ac.id
3   Systems Ecology and Resource Management, Department of Organism Biology, Université Libre de Bruxelles (ULB), Avenue F.D. Roosevelt 50, CPi 264/1, 1050 Brussels, Belgium; koedamnico@gmail.com
4   Marine Biology Research Group, Universiteit Gent, Krijgslaan 281—S8, 9000 Gent, Belgium
5   Centre for Environmental Sciences (CMK), Universiteit Hasselt, Agoralaan z/n, 3590 Diepenbeek, Belgium
6   Mangrove Specialist Group (MSG), Species Survival Commission (SSC), International Union for the Conservation of Nature (IUCN), 1196 Gland, Switzerland
7   Department of Aquatic Resources Management, Universitas Bangka Belitung, Bangka 33172, Indonesia; rizzamuftiadi@ubb.ac.id
*   Correspondence: sucipuspita.sari@ugent.be

**Abstract:** The current state of the mangrove ecosystem on Bangka Island requires urgent attention from the local government to protect, restore, and conserve the remaining mangrove areas. Hence, this study endeavors to assess the species composition of mangroves on Bangka Island, examining their correlation with edaphic factors and shedding light on the zonation pattern within the region. We examined species composition, edaphic factors, and zonation patterns along 20 m × 100 m transects perpendicular to the waterfront at 22 sampling sites distributed across Bangka Island. Our findings revealed the presence of 21 mangrove species from ten families, including two mangrove associates. Among the documented species, the *Rhizophoraceae* family exhibited the highest floristic abundance with nine species. Edaphic factors (soil texture, pore-water salinity, N-total, P, and K) significantly influenced mangrove species composition ($p < 0.05$). However, these factors explained only 37.2% of the overall variability, suggesting additional factors contribute to the diverse zonation and composition of mangroves on Bangka Island. This study has relevant implications for the conservation and management of mangroves on Bangka Island. By gaining insight into the specific site's floristic composition, overall richness, and distribution, our findings can guide effective conservation and restoration strategies by understanding the factors shaping mangrove composition.

**Keywords:** canonical correspondence analysis; mangrove biodiversity; mangrove ecology; species richness; tin mining

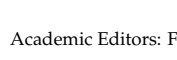

## 1. Introduction

Mangroves comprise a collection of trees and shrubs that are typically situated in the estuaries and coastal regions of tropical and subtropical areas worldwide [1–4]. Mangroves represent a significant ecological and economic asset by offering a biotope with crucial habitat [1,5]. They generate an annual ecosystem service value of approximately 21,100 USD/ha.a [6–8], contributing significantly to coastal livelihoods on a global scale [9,10]. Despite their ecological and economic importance, there has been a net loss of approximately 3.4% (5245 km²) in the global extent of mangroves over 24 years from 1996 to 2020 [9]. Inspiringly, there has been a global decline in mangrove deforestation rates, ranging from 0.16% to 0.39% per year [10,11]. Nonetheless, the progress in curbing this

decline is highly uneven, with some areas successfully halting the loss while others, such as Myanmar and Malaysia, continue to experience high rates [10].

Indonesia has the most considerable mangrove extent and the highest mangrove tree species richness in the world [2,12–14], containing approximately 24% of the world's mangroves [15] or around 3,364,080 ha [16]. However, it is also among the countries with the highest rates of mangrove loss [13,17,18], with an estimated 1 million ha lost since 1800 [12]. Human activities have adversely affected Mangroves in Indonesia, resulting in the loss of 182,091 ha through deforestation and the degradation of 79,050 ha between 2009 and 2019 [15,19]. These activities include aquaculture, agriculture, timber harvesting, oil palm plantation, land conversion, mining, and logging [12,14,19,20]. The highest peak of mangrove conversion to aquaculture (56,984 ha) occurred during 2015–2018 and was linked to increased deforestation emissions [15]. One of the regions in Indonesia that suffered is Bangka Island, which belongs to the Bangka Belitung Province.

Thus, the mangrove ecosystem in Indonesia requires urgent attention from competent authorities to protect, restore, and conserve the remaining mangrove areas. Effective policy interventions can address most of the drivers of mangrove loss [21]. To ensure the sustainability of the mangrove ecosystems, all stakeholders at all levels, as outlined in Presidential Decree No. 73 of 2012, must support these actions [22]. Indonesia has expressed a strong commitment to reducing national emissions by 31.89% and 43.20% under unconditional and conditional mitigation scenarios by 2030, as well as to protecting and restoring mangroves [22,23]. The target is to restore 3.49 million hectares of mangroves by 2045, as outlined in the Coordinating Ministry of Economy Regulation No. 4 of 2017. Additionally, Presidential Decree No. 120 of 2020 endeavors to restore mangroves on 600,000 hectares by 2024, contributing to improving community welfare in various provinces, including Bangka Island.

Over the past decade, approximately 767.33 ha of mangrove resources have been lost on Bangka Island due to coastal erosion or anthropogenic activities [9]. The local communities have utilised mangroves for firewood, charcoal, and building materials, as well as converted mangrove areas into ponds and settlements, contributing to a decreasing mangrove extent in various parts of the island [24–27]. Aside from the substantial coverage of mangrove forests, this region is the second-largest global tin producer, with a mining history [28] since the 17th century [29]. Mining operations have adversely affected the environment, including mangrove ecosystems. These activities have led to environmental degradation characterised by compromised water quality, pollution, and sedimentation [27,28,30,31]. Open-pit mining is one of the tin extraction techniques utilised in the Bangka Islands [32]. This method involves the use of water guns and gravel pumps to extract tin sand from land surfaces or alluvial deposits [33]. Alluvial tin deposits, specifically with cassiterite ($SnO_2$), are exposed by removing vegetation and non-tin deposit overburden [34]. However, removing topsoil during mining operations has significant consequences, such as eliminating vegetation, altering its structure and composition, changing topography, reducing wildlife habitat, and modifying land use [34–36]. These impacts include soil structure deterioration, shifts in soil texture (70% to 97% sand fraction), the loss of organic matter, and decreased soil fertility [34,37,38]. As part of the extraction process, the tin sand is washed, releasing effluent that flows into nearby rivers and coastal areas. This process leads to sedimentation and water contamination, impacting associated ecosystems.

The current state of mangrove species diversity presents an immense challenge due to a lack of current scientific knowledge and long-term data to predict their responses to human impacts [1]. The urgency lies in the imperative to enhance country-level data on mangroves' extent, health, and ecosystem service provision [3]. Therefore, acquiring detailed information through a comprehensive inventory initiative is crucial for informing conservation and restoration strategies. Despite a multitude of studies addressing mangrove distribution and community structure on Bangka Island [24–27,30,31,39–46], scientific data on species distribution, edaphic factors, and the potential zonation pattern of site-specific mangroves, specifically in the wider geographic area of Bangka Island,

are mostly lacking or fragmented. A comprehensive study of the mangrove community across Bangka Island is needed to gain insights into the distribution patterns of various species. Such insights are essential for assessing the biodiversity of mangrove forests and enhancing the effectiveness of conservation and restoration efforts by ensuring the selection of appropriate sites with representative species composition and site characteristics.

In this study, we examined 22 sites across Bangka Island that were chosen to represent the diverse range of mangrove ecosystems in terms of typology and coverage area. We measured the composition of mangroves and the edaphic factors, including soil texture, pore-water salinity, N-total, P, and K, within a 20 m × 100 m transect perpendicular to the waterfront in each sampling site. We aimed to (I) determine whether there are differences in the composition and diversity of mangroves between sampling sites, (II) determine the general pattern of mangrove zonation along transects perpendicular to the water line on Bangka Island, if present, and (III) investigate the relationship between mangrove composition and edaphic factors. This study wishes to contribute to a comprehensive view of the mangrove conditions on Bangka Island. By addressing the lack of detailed data on species distribution and site characteristics, this study provides a foundational framework for future studies and management plans. Furthermore, this baseline study is expected to contribute to developing effective monitoring and conservation strategies on Bangka Island.

## 2. Materials and Methods

### 2.1. Study Area

This study was carried out in the mangrove areas of Bangka Island (Indonesia), located between the coordinates 104°50′ to 107°08′ E and 0°50′ to 2°30′ S. The island has an area of 11,623.54 km$^2$, bordered by two channels to the east and its west (Gaspar Strait and Bangka Strait, respectively), the Java Sea to the south, and the Natuna Sea to the north [47]. The climate in the region is tropical with two seasons, dry from June to October and rainy from September to May. According to the Köppen climate classification, the climate type is equatorial fully humid (Af) [48], with an average humidity of 82.3%, and the atmospheric pressure is an average of 1007.9 mbar. The average air temperature reaches 28.2 °C. The average precipitation is around 3012.9 mm, with a number of 234 days of rain [47]. The island is situated in the Sunda Shelf/Java Sea ecoregion [49] and is not subjected to tropical cyclones [50]. The tidal amplitude is between 2 m and 3 m [51]. Sea level changes in Indonesia are reported to be +4.5 mm/year over a period of 25 years [52].

### 2.2. Data Collection and Analysis

Field surveys and environmental assessments were conducted from July to August 2021. Mangrove sampling locations were determined based on the mangrove extent map, mangrove sites in the reference article [39], human activity near the forests, and accessibility combined (Figure A1). A total of 22 mangrove sites were sampled all around Bangka Island (Table A1) in 4 regencies (Bangka Barat, Bangka Tengah, Bangka Selatan, and Bangka). The sampling design used to collect the mangrove data and soil sample was adapted from the Biomass assessment protocol used for a mangrove forest in Suriname [53]. A transect line of 100 m was laid perpendicular to the waterfront in each sampling site, in which a rectangular plot of 100 m × 20 m was established. Mangrove observations were performed in every 10 m × 10 m plot along the line transect. The total observed area in each site was 20 plots, hence 2000 m$^2$. Sampling location coordinates were marked using GPS Garmin 76csx (Estimated Position Error (EPE) ranged from 4 to 7 m with a mean of 4.6 m).

All mangroves' stems within each plot were identified up to the species level. Mangrove species were identified using identification sources and keys [54–56], and the nomenclature was confirmed using the International Plant Name Index (IPNI) through the website (https://www.ipni.org/ accessed on 3 July 2023). Roots, leaves, flowers, fruits, and propagules were all captured to support identification, if present. The girth at breast height (GBH at 1.3 m above ground level or just above the buttress) of all stems was recorded in each plot.

Occurrences of *Acanthus ilicifolius* L., *Acrostichum speciosum* Wild., *Nypa fruticans* Wurmb, and *Pandanus tectorius* Parkinson were recorded. However, we did not include their GBH in the data analysis because these plant life forms do not enable relevant measurements to be taken and hence there are no data included for these species regarding their abundance in the analysis. Epiphytes and parasites inside the plots were not recorded. Soil samples were taken at a 0–100 cm depth using a stainless-steel D-section corer, and a minimum of 500 g was collected. Soil samples from each site were analysed at PT. Soil samples from each site were analysed at PT. Global Quality Analytical (Bogor, Indonesia). The soil texture was determined using the Pipette method, while the nutrient concentrations were assessed using Spectrophotometric and ICP techniques. The pore-water salinity was measured at the site (in situ) with a refractometer [57]. The mangrove community structure was evaluated to determine variables such as tree density, basal area, relative density, relative frequency, relative dominance, and importance value. The importance value was determined by adding the relative value of density, frequency, and dominance [58].

*2.3. Statistical Analysis*

Groupings of species and assemblage patterns were defined based on Bray–Curtis similarity using multivariate methods and implemented in PRIMER v6 (Plymouth Routines in Multivariate Ecological Research; Playmouth Marine Laboratory, UK) [59]. A dendrogram and Multi-Dimensional Scaling (MDS) plot displayed the categorised group. The Shannon index (H′), Analysis of Similarity (ANOSIM) and Similarity percentage (SIMPER) were also calculated in PRIMER v6 to determine species diversity and whether there were significant differences in clustering. SIMPER analysis helped identify the species responsible for the observed differences. The type of presence/absence and square root transformation of the mangrove species were used for the analysis. The relationship between the mangrove species and edaphic factors was analysed using Canonical Correspondence Analysis (CCA) in the software Canoco 5 (Microcomputer Power; USA/NY). Planet & NICFI Basemaps for Tropical Forest Monitoring—Tropical Asia (source: https://developers.google.com/earth-engine/datasets/catalog/projects_planet-nicfi_assets_basemaps_asia accessed on 3 July 2023).

**3. Results**

The sample area covered 22 sites (Table A1) in four different parts of Bangka Island: Bangka Barat, Bangka Tengah, Bangka Selatan, and Bangka. This section divides the results into four distinct parts: (1) mangrove species composition and distribution on Bangka Island, (2) zonation pattern along the observation transect, (3) mangrove diversity, and (4) the relationship between edaphic factors and mangrove abundance.

*3.1. Mangrove Species Composition and Distribution on Bangka Island*

To identify the various species of vegetation present in each of the sampling sites on Bangka Island, we analysed the characteristics of the vegetation using identification keys and descriptions, as given in the methods section. We found 21 mangroves species (Table 1) belonging to ten families and two mangrove associates (sensu Kitamura et al., 1997) [54]. This number, combined with previous studies on Bangka Island (Table 2), covers around 70% of the species identified in Indonesia [60]. The highest numbers of species were recorded at Merbau Beach and Dante Island, with 12 species each, while the lowest numbers of species were recorded at Kota Kapur and Sukal, with only 1 each. Among all species, one species found (*Avicennia lanata*) has a Vulnerable (VU) status, and one other species (*Ceriops decandra*) is Near Threatened (NT), according to the IUCN global Red List of Threatened Species (Table 1). In addition, at the family level, *Rhizophoraceae* had the highest number of species and was the most common mangrove family in the area. At least one species (*Rhizophora apiculata*) from the family *Rhizophoraceae* was documented in five sites (Pangkul Beach, Tanjung Bajun, Tanjung Punai, Tanjung Pura, and Tanjung Sunur). Furthermore, a maximum of six species (*Bruguiera gymnorrhiza*, *Bruguiera sexangula*, *Ceriops*

*decandra*, *Ceriops tagal*, *Rhizophora apiculata*, and *Rhizophora mucronata*) from this family were noted in one location (Dante Island). This indicates that the *Rhizophoraceae* family was floristically the most abundant.

**Table 1.** IUCN Red List status of mangrove species identified on the Bangka Island sampling sites (incl. mangrove associates).

| Species | Family | IUCN Red List |
|---|---|---|
| True mangrove | | |
| *Acrostichum speciosum* Wild. | Pteridaceae | LC |
| *Aegiceras corniculatum* (L.) Blanco | Primulaceae | LC↓ |
| *Avicennia alba* Blume | Acanthaceae | LC↓ |
| *Avicennia lanata* Ridl. | Acanthaceae | VU |
| *Bruguiera cylindrica* (L.) Blume | Rhizophoraceae | LC↓ |
| *Bruguiera gymnorrhiza* (L.) Lamk. | Rhizophoraceae | LC↓ |
| *Bruguiera parviflora* (Roxb.) Wight & Arn. ex Griff. | Rhizophoraceae | LC↓ |
| *Bruguiera sexangula* (Lour.) Poir. | Rhizophoraceae | LC↓ |
| *Ceriops decandra* (Griff.) Ding Hou | Rhizophoraceae | NT↓ |
| *Ceriops tagal* (Perr.) C.B. Robinson | Rhizophoraceae | LC↓ |
| *Excoecaria agallocha* L. | Euphorbiaceae | LC↓ |
| *Lumnitzera littorea* (Jack) Voigt. | Combretaceae | LC↓ |
| *Lumnitzera racemosa* Willd. | Combretaceae | LC↓ |
| *Nypa fruticans* Wurmb | Arecaceae | LC? |
| *Rhizophora apiculata* Blume | Rhizophoraceae | LC↓ |
| *Rhizophora mucronata* Lamk. | Rhizophoraceae | LC↓ |
| *Rhizophora stylosa* Griff. | Rhizophoraceae | LC↓ |
| *Scyphiphora hydrophyllacea* C.F.Gaertn. | Rubiaceae | LC↓ |
| *Sonneratia alba* J. Smith | Lythraceae | LC↓ |
| *Sonneratia caseolaris* (L.) Engl. | Lythraceae | LC↓ |
| *Xylocarpus granatum* J.Koenig | Meliaceae | LC↓ |
| Mangrove Associate | | |
| *Acanthus ilicifolius* L. | Acanthaceae | LC? |
| *Pandanus tectorius* Parkinson | Pandanaceae | LC? |

IUCN Red List Status: LC = Least Concern, NT = Near Threatened, VU = Vulnerable. Downward arrow indicates decreasing, question mark indicates unknown. Sources report the vulnerable or threatened status at a global level.

**Table 2.** Identified mangrove species on Bangka Island from various research reports (incl. mangrove associates).

| Species | 1 | 2 | 3 | 4 | 5 | 6 | 7 | 8 | 9 | 10 | 11 | 12 | 13 | 14 | 15 |
|---|---|---|---|---|---|---|---|---|---|---|---|---|---|---|---|
| *Acanthus ilicifolius* | ● | | | | | | | ● | | | | ● | | | |
| *Acrostichum aureum* L. | ◇ | | | ● | | | ● | | | | | | | | |
| *Acrostichum speciosum* | ● | | | | | | | | | ● | | ● | ● | | |
| *Aegiceras corniculatum* | ● | | | ● | | | | | | | | | | | |
| *Avicennia alba* | ● | ● | | | | ● | | | | | | | ● | | |
| *Avicennia lanata* | ● | | | ● | | ● | | | | | ● | | | | |
| *Avicennia marina* | ◇ | | | | ● | | | ● | | | ● | ● | | | |
| *Bruguiera cylindrica* | ● | | | ● | | | | | | | | | | | |
| *Bruguiera gymnorrhiza* | ● | ● | ● | ● | | | ● | ● | | ● | | ● | | ● | |
| *Bruguiera hainesii* | | | | | ● | | | | | | | | | | |
| *Bruguiera parviflora* | ● | | | ● | | | | | | ● | | | | | |
| *Bruguiera sexangula* | ● | | | ● | | | | | | | | | ● | ● | |
| *Ceriops decandra* | ● | | | ● | | | | | | | | | | ● | |
| *Ceriops tagal* | ● | ● | ● | ● | | | ● | | | | | | | ● | |
| *Cycas circinalis* L. | | | | | | | | | ● | | | | | | |
| *Dillenia indica* L. | | | | | | | | | ● | | | | | | |
| *Excoecaria agallocha* | ● | | ● | ● | | | | | | | | | ● | | |
| *Heritiera littoralis* Aiton | ◇ | | | | | | | | | | | | ● | | |

**Table 2.** *Cont.*

| Species | 1 | 2 | 3 | 4 | 5 | 6 | 7 | 8 | 9 | 10 | 11 | 12 | 13 | 14 | 15 |
|---|---|---|---|---|---|---|---|---|---|---|---|---|---|---|---|
| *Hibiscus tiliaceus* L. | | | | | | | | | • | | | | | | |
| *Lumnitzera littorea* | • | | • | • | | | • | • | | | | • | | | |
| *Lumnitzera racemosa* | • | | | | | | • | | • | | | | | | |
| *Nypa fruticans* | • | | | • | | | | | • | | • | • | • | | |
| *Pandanus tectorius* | • | | | | | | | | • | | | | | | |
| *Pemphis acidula* J.R.Forst. & G.Forst. | ◊ | | • | | | | | | | | | | | | |
| *Pongamia pinnata* (L.) Merr. | | | • | | | | | | | | | | | | |
| *Rhizophora apiculata* | • | | • | • | | • | • | • | • | • | • | • | • | • | • |
| *Rhizophora lamarckii* | | | | • | | | | | | • | | | | | |
| *Rhizophora mucronata* | • | • | • | • | • | • | | • | • | • | | • | • | • | • |
| *Rhizophora stylosa* | • | | • | • | | | | | | • | | | | | |
| *Scyphiphora hydrophyllacea* | • | | | • | | | | | | | | | | | |
| *Sesuvium portulacastrum* (L.) L. | | | | • | | | | | | | | | | | |
| *Sonneratia alba* | • | | • | • | • | | • | • | • | • | • | • | | • | |
| *Sonneratia caseolaris* | • | | | • | | • | | | • | | | | | | |
| *Sonneratia ovata* | | | | • | | | | | | | | | | | |
| *Talipariti tiliaceum* (L.) Fryxell | | | | • | | | | | | | | | | | |
| *Thespesia populnea* (L.) Sol. ex Corrêa | | | • | | | | | | | | | | | | |
| *Xylocarpus granatum* | • | | | • | | | • | | | • | | | • | | • |
| *Xylocarpus moluccensis* (Lam.) M.Roem. | | | | | | | | | • | | | | | | |
| Total | 27 | 4 | 12 | 22 | 4 | 5 | 8 | 7 | 11 | 9 | 5 | 9 | 10 | 7 | 3 |
| Study area | f | a | a | b | c | d | a | c | c | e | c | c | c | d | a |

Notes: • (Presence); ◊ (Presence outside the observation plots). 1. Present study; 2. Affressia et al. (2017) [61]; 3. Akhrianti (2019) [62]; 4. Akhrianti and Gustomi (2021) [46]; 5. Farhaby (2019) [30]; 6. Farhaby and Anwar (2022) [27]; 7. Farhaby et al. (2022) [63]; 8. Fazlina et al. (2021) [64]; 9. Henri et al. (2022) [31]; 10. Rosalina and Rombe (2021) [26]; 11. Savira et al. (2018) [24]; 12. Susi et al. (2018) [42]; 13. Zulia et al. (2019) [43]; 14. Farhaby et al. (2020) [44]; 15. Farhaby et al. (2020) [45]; a. Bangka Selatan; b. Pangkalpinang; c. Bangka Tengah; d. Bangka; e. Bangka Barat; f. Bangka Island.

### 3.1.1. Presence/Absence

Multivariate analysis was conducted to examine the similarities observed in the mangrove floristic composition among the sampling sites based on species presence/absence. The analysis included cluster and ordination techniques. According to the cluster analysis results (Bray–Curtis similarity) and corresponding Multi-Dimensional Scaling (MDS) plot with around 45% similarity, the mangrove species composition within the sampling sites could be divided into two groups (Figure 1). Group (A) consisted of Batu Beriga, Kota Kapur, Pangkul Beach, Sampur Beach, Sukal, Tanjung Bajun, Tanjung Labu, Tanjung Punai, Tanjung Pura, and Tanjung Sunur. Group (B) consisted of Dante Island, Ibul Island, Jebu Laut, Kepoh, Lubuk Besar, Merbau Beach, Panjang Island, Pejem Beach, Sadai, Seniur Island, Tanjung Pao, and Tanjung Tada. However, three groups exhibited a 100% similarity in terms of the species composition of their presence/absence. Group (X) was Batu Beriga and Tanjung Labu; (Y) was Tanjung Punai, Tanjung Pura, and Tanjung Sunur; and (Z) was Sukal and Kota Kapur. For instance, the mangrove species composition in Group (Z) consisted of *Avicennia alba*, *Rhizophora apiculata*, and *Sonneratia alba*. This observation indicates that the formation of these groups is attributed to the similarity in the presence/absence of species within each group.

To assess whether there were significant differences in the mangrove species' composition between the two groups based on their presence/absence (Group A and B), we conducted an Analysis of Similarity (ANOSIM) and Similarity Percentage (SIMPER). The results showed a significant difference between Group A and B ($p = 0.0001$ and Global R = 0.89). The average dissimilarity in the mangrove species presence between the two groups was 72%, with *Bruguiera gymnorrhiza* being the major contributor (14%). Within sites in Group A, the average similarity in the mangrove species presence was 62%, with three species contributing to the similarity and *Sonneratia alba* being the major contributor (60%). On the other hand, within sites in Group B, the average similarity in the mangrove

species composition was 72%, with eight species contributing to the similarity. *Bruguiera gymnorrhiza* and *Rhizophora apiculata* accounted for an equal proportion of 24% in the species composition similarity.

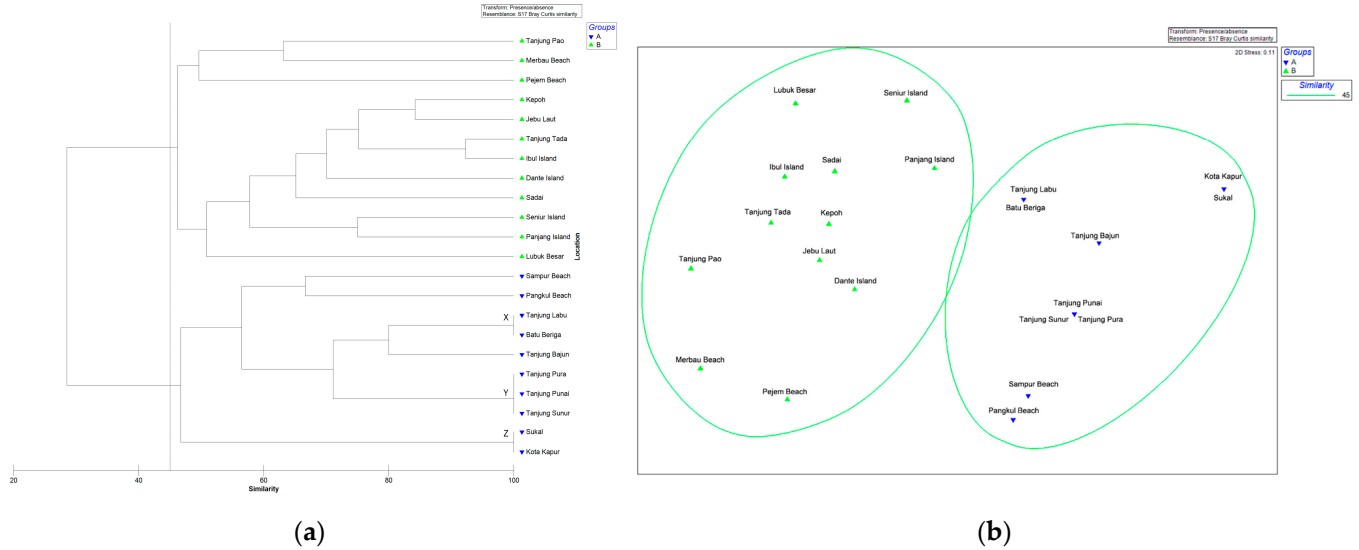

(**a**)        (**b**)

**Figure 1.** Visualizing mangrove distribution on Bangka Island: (**a**) Dendrogram of the similarity between study sites based on the presence/absence of mangroves; (**b**) MDS configuration for the same locations (stress 0.11).

### 3.1.2. Abundance

We conducted a multivariate analysis to examine the similarities in the mangrove composition among the sampling sites based on species abundance, using the same method as the presence/absence of species. This additional analysis aimed to support the clustering results obtained from the presence/absence data. Notably, the analysis excludes *Acanthus ilicifolius*, *Acrostichum speciosum*, *Nypa fruticans*, and *Pandanus tectorius*. The multivariate analysis for abundance only considered true mangrove species at the tree stage. The results revealed that the mangrove species composition within sampling sites could be categorised into five distinct groups based on 40% similarity (Figure 2). Group (I) was Dante Island, Ibul Island, Jebu Laut, Kepoh, Merbau Beach, Sadai, Tanjung Pao, and Tanjung Tada; Group (II) was Batu Beriga, Pejem Beach, Sampur Beach, Tanjung Bajun, Tanjung Labu, Tanjung Pura, and Tanjung Sunur; Group (III) was Panjang and Seniur Island; (IV) was Kota Kapur, Pangkul Beach, Sukal, and Tanjung Punai; and Group (V) was Lubuk Besar. This suggests that the grouping is influenced by the similarity in the species abundance composition within each group.

We also performed an ANOSIM and SIMPER analysis to compare the mangrove species composition among the five groups based on their abundance. The results showed that each group had a distinct average similarity in their mangrove species composition ($p = 0.0001$ and Global R = 0.88). The similarity within sites in Group (I) and Group (II) majorly was contributed by *Rhizophora apiculata*, with average similarities of 38% and 77%, respectively. Group (III) had an average similarity of 59%, with *Rhizophora mucronata* being the major contributor (84%), and Group (IV) had an average similarity of 68%, with *Sonneratia alba* being the major contributor (95%).

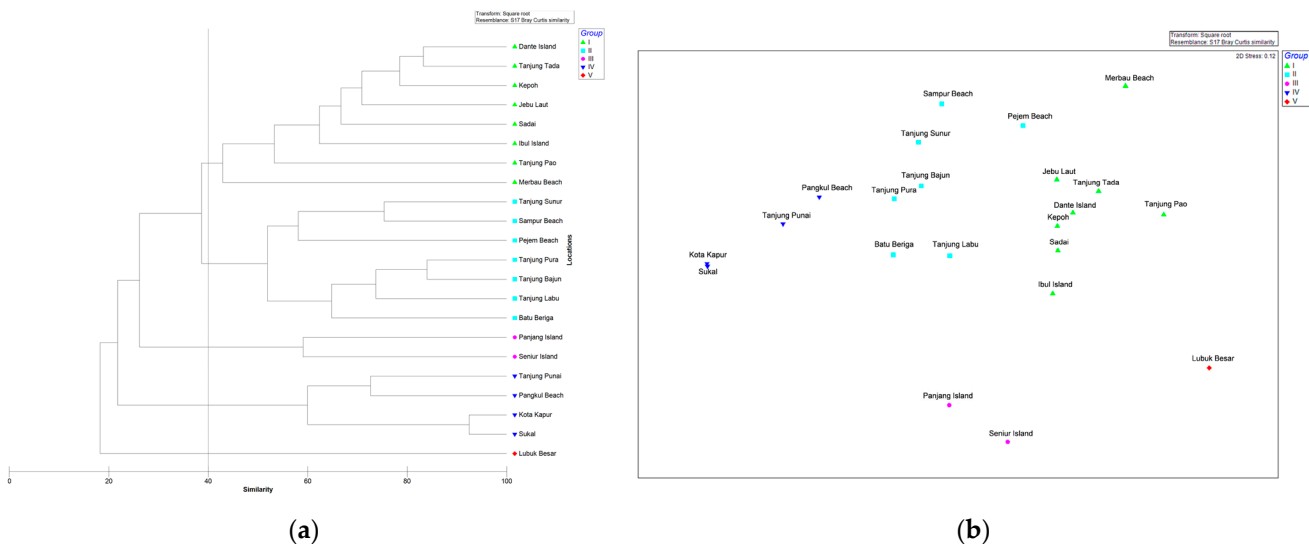

**(a)**                                        **(b)**

**Figure 2.** Visualizing mangrove distribution on Bangka Island: (**a**) Dendrogram of the similarity between study sites based on the abundance of mangroves; (**b**) MDS configuration for the same locations (stress 0.12).

### 3.2. Zonation Pattern along the Observation Transect

We surveyed the species' presence along the transect to analyse the zonation patterns in the selected mangrove survey locations. We observed *Rhizophora apiculata* and *Sonneratia alba* in nearly all sites. At Sukal and Kota Kapur, where mono-species formations were found, *Sonneratia alba* was present along the entire 100 m transect. On the other hand, in locations with numerous species, such as Dante Island, the seaward-facing zone was covered by *Rhizophora apiculata*, which was more abundant within 0–50 m. The middle to landward zone was covered by *Bruguiera gymnorrhiza* and *Bruguiera sexangula*, mixed with *Aegiceras corniculatum*, *Avicennia alba*, *Ceriops decandra*, *Ceriops tagal*, *Excoecaria agallocha*, *Sonneratia alba*, and *Xylocarpus granatum*. In addition, *Acrostichum speciosum* covered the forest floor in approximately 5% to 75% of the observed plots.

To identify a possible zonation pattern, we performed segmentation analysis on planet satellite images (5 m spatial resolution) [65]. Our segmentation analysis strongly suggested the presence of mangrove zones generally parallel to the coastline, and also in wide forests like Kota Kapur, which was established as monospecific in our sampling. This indicates the possible presence of another species beyond the 100 m transect, in which only *Sonneratia alba* was recorded. It emphasises the need for additional surveys in the widest mangrove forests. Conversely, in heterogenous species areas like Dante Island, the mangrove zone displayed a mosaic pattern with weak zonation (Figure 3). These findings suggest varying zonation patterns on Bangka Island, with a potential ecological gradient observed in apparently homogenous areas and a mosaic zonation pattern in heterogenous areas.

### 3.3. Diversity of True Mangrove Species

The Shannon Index was used to determine species diversity, which ranged from 0.00 (Sukal and Kota Kapur) to 1.79 (Merbau Beach). Despite this, according to the Shannon Index, all sites had relatively low mangrove species diversity. The total tree density varied from 460 trees ha$^{-1}$ (Tanjung Bajun) to 2500 trees ha$^{-1}$ (Sadai) across all sites. *Rhizophora apiculata* had the highest importance value (IV) in ten sites. These sites include Pejem Beach, Tanjung Sunur, Sadai, Tanjung Bajun, Tanjung Labu, Jebu Laut, Sampur Beach, Kepoh, Tanjung Tada, and Merbau Beach. The highest IV of *Rhizophora apiculata* was observed in Pejem Beach (234.51%). Meanwhile, *Lumnitzera racemosa* was recorded as the dominant species on Lubuk Besar only (143.20%) (Table 3). Overall, the diversity and tree density

on Bangka Island were slightly higher than Papua [66], which has a higher number of mangrove species (Table 4).

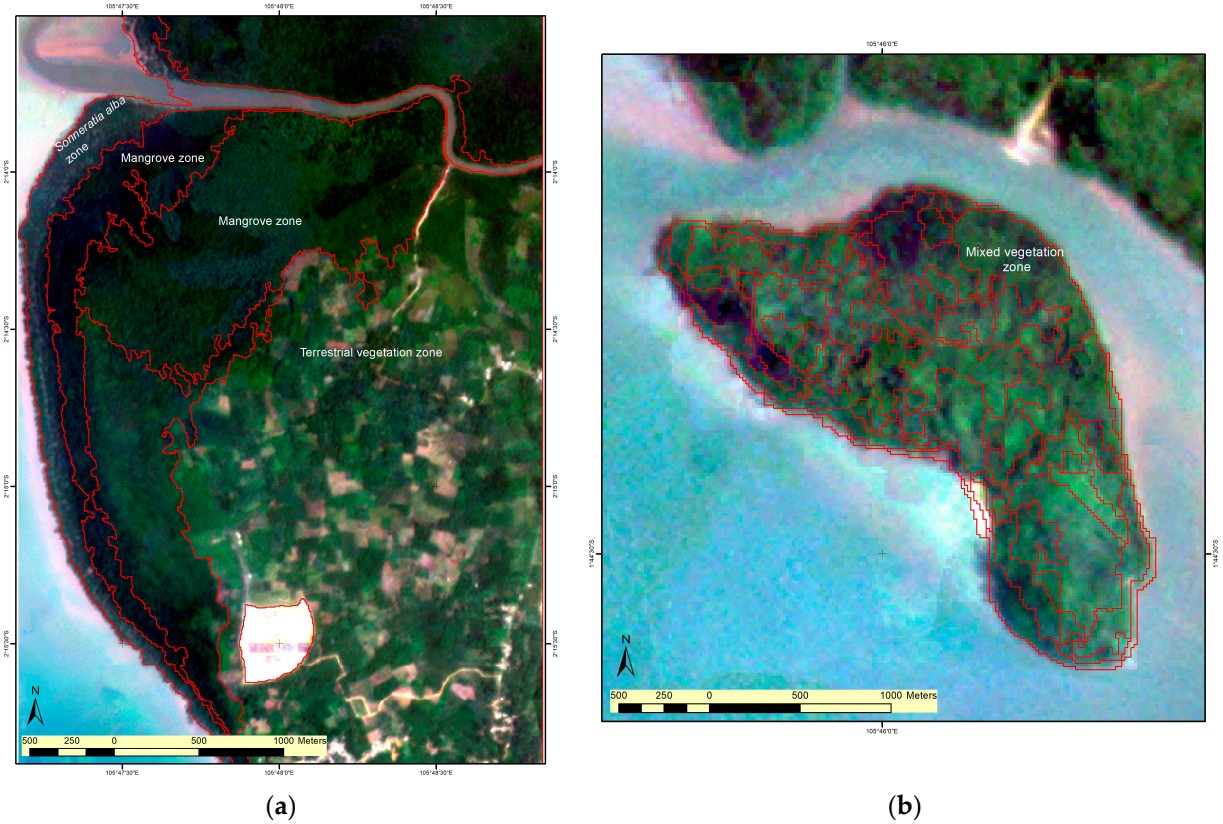

(**a**)                    (**b**)

**Figure 3.** Example of a segmentation analysis for a mangrove zonation pattern; the red line indicates the tentative boundaries between zones: (**a**) Kota Kapur; (**b**) Dante Island.

**Table 3.** Tree density, number of species, Shannon Index (H'), and importance value (IV) of dominant mangrove species at each sampling site, listed in alphabetical order of location.

| Location | Number of Species | Tree Density (Tree ha$^{-1}$) | H' | Dominant Species | RD (%) | RF (%) | RDo (%) | IV (%) |
|---|---|---|---|---|---|---|---|---|
| Batu Beriga | 3 | 1760 | 0.89 | *Sonneratia alba* | 50.85 | 48.15 | 42.89 | 141.89 |
| Dante Island | 11 | 1500 | 1.73 | *Xylocarpus granatum* | 16.33 | 22.39 | 38.75 | 77.47 |
| Ibul Island | 6 | 1870 | 1.07 | *Rhizophora mucronata* | 51.60 | 32.73 | 47.37 | 131.70 |
| Jebu Laut | 9 | 2115 | 1.07 | *Rhizophora apiculata* | 69.74 | 32.20 | 47.53 | 149.47 |
| Kepoh | 9 | 1675 | 1.48 | *Rhizophora apiculata* | 40.00 | 32.56 | 41.51 | 114.07 |
| Kota Kapur | 1 | 805 | 0.00 | *Sonneratia alba* | 100 | 100 | 100 | 300 |
| Lubuk Besar | 10 | 920 | 1.24 | *Lumnitzera racemosa* | 66.30 | 34.38 | 42.52 | 143.20 |
| Merbau Beach | 9 | 1675 | 1.79 | *Rhizophora apiculata* | 25.07 | 25.00 | 23.27 | 73.34 |
| Pangkul Beach | 6 | 1880 | 0.56 | *Sonneratia alba* | 84.31 | 60.00 | 93.87 | 238.18 |
| Panjang Island | 5 | 1240 | 0.94 | *Rhizophora mucronata* | 65.32 | 50.00 | 40.11 | 155.44 |
| Pejem Beach | 6 | 1435 | 0.47 | *Rhizophora apiculata* | 87.46 | 59.38 | 87.67 | 234.51 |
| Sadai | 9 | 2500 | 1.17 | *Rhizophora apiculata* | 65.40 | 35.71 | 77.74 | 178.86 |
| Sampur Beach | 6 | 1965 | 1.21 | *Rhizophora apiculata* | 59.03 | 35.56 | 53.87 | 148.46 |
| Seniur Island | 3 | 1740 | 0.18 | *Rhizophora mucronata* | 95.98 | 76.92 | 96.72 | 269.62 |

**Table 3.** *Cont.*

| Location | Number of Species | Tree Density (Tree ha$^{-1}$) | H' | Dominant Species | RD (%) | RF (%) | RDo (%) | IV (%) |
|---|---|---|---|---|---|---|---|---|
| Sukal | 1 | 595 | 0.00 | *Sonneratia alba* | 100 | 100 | 100 | 300 |
| Tanjung Bajun | 2 | 460 | 0.30 | *Rhizophora apiculata* | 91.30 | 60 | 25.67 | 176.97 |
| Tanjung Labu | 3 | 675 | 0.59 | *Rhizophora apiculata* | 80.74 | 50.00 | 26.67 | 157.41 |
| Tanjung Pao | 6 | 1525 | 1.33 | *Ceriops tagal* | 45.57 | 28.57 | 31.29 | 105.44 |
| Tanjung Punai | 3 | 1290 | 0.60 | *Sonneratia alba* | 81.40 | 57.14 | 86.57 | 225.11 |
| Tanjung Pura | 3 | 600 | 0.67 | *Sonneratia alba* | 31.67 | 32.00 | 87.00 | 150.66 |
| Tanjung Sunur | 3 | 1590 | 0.63 | *Rhizophora apiculata* | 80.19 | 51.61 | 57.01 | 188.81 |
| Tanjung Tada | 7 | 1520 | 1.32 | *Rhizophora apiculata* | 35.20 | 25.49 | 41.49 | 102.18 |

Notes: H': Shannon Index; RD: Relative Density; RF: Relative Frequency; RDo: Relative Dominance; IV: Importance Value.

**Table 4.** Comparison of mangroves on Bangka Island with other mangrove forests.

| No. | Research Area | Country | Number of Species | Dominant Species | H' | Tree Density (Trees ha$^{-1}$) | Observation Area/Site (m$^2$) |
|---|---|---|---|---|---|---|---|
| 1 | Mimika District, Papua | Indonesia | 66 | *R. apiculata, B. gymnorrhiza* and *A. marina* | 0.62–1.19 | 577–1345 | 900 |
| 2 | Kayan-Sembakung Delta, North Kalimantan | Indonesia | 9 | *R. Apiculata* and *Sonneratia* sp. | <2 | 22–1635 | NA |
| 3 | Gulf of Uraba, Colombian Caribbean | South America | 13 | *R. Mangle* and *A. germinans* | NA | 166–1763 | 500 |
| 4 | Vidattaltivu | Sri Lanka | 7 | *A. marina* | 0–1.596 | 800–5100 | 100 |
| 5 | Kenya and Sri Lanka | Kenya and Sri Lanka | 13 | *C. tagal, R. mucronata, E. agallocha* | NA | 892–1915 | 100 |
| 6 | Coringa, Kakinada Bay | India | 15 | *A. marina* | NA | 470–17,310 | NA |
| 7 | Tumpat, Kelantan Delta | Malaysia | 5 | *S. caseolaris* | NA | 790–1320 | NA |
| 8 | Cochin, Kerala | India | 14 | *A. officinalis* | 1.595–2.66 | 7680–11,760 | NA |
| 9 | Sibuti, Sarawak | Malaysia | 9 | *R. apiculata, X. granatum,* and *N. fruticans* | 1.18 | 1500–2340 | 1300 |
| 10 | Air Telang Protected Forest, South Sumatra | Indonesia | 20 | *N. fruticans, R. apiculata,* and *A. aureum* | 0.00–0.73 | 0–766.667 | 1000 |
| 11 | Pulau Rimau Protection Forest, Banyuasin | Indonesia | 15 | *N. fruticans* and *S. caseolaris* | NA | NA | 1200 |
| 12 | Belitung Island | Indonesia | 24 | *Bruguiera* and *Rhizophora* spp. | 2.19–2.7 | NA | NA |
| 13 | Gulf of Khambhat, Gujarat | India | 16 | *A. marina* | 0–1.179 | 777–97,222 | 90 |
| 14 | Andaman Island | India | 28 | *Rhizophora* sp. | NA | 133–14,000 | 2000 |
| 15 | Kerala | India | 18 | *A. officinalis* | 1.44–3.75 | 10–13,846 | 500 |
| 16 | Bangka Island | Indonesia | 23 | *R. apiculata* | 0.00–1.79 | 460–2500 | 2000 |

Notes: NA (Not Available); 1. Setyadi et al. (2021) [66]; 2. Seftianingrum et al. (2020) [67]; 3. Urrego et al. (2014) [68]; 4. Cooray et al. (2021) [69]; 5. Dahdouh-Guebas et al. (2002) [70]; 6. Satyanarayana et al. (2002) [71]; 7. Satyanarayana et al. (2010) [72]; 8. Rani et al. (2018) [73]; 9. Shah et al. (2016) [74]; 10. Eddy et al. (2019) [75]; 11. Yuliana et al. (2019) [76]; 12. Irawan et al. (2021) [77]; 13. Singh (2020) [78]; 14. Sreelekshmi et al. (2020) [79]; 15. Sreelekshmi et al. (2018) [80]; 16. Present study.

### 3.4. The Relation between Edaphic Factors and Mangrove Abundance

The observed soil pore-water salinity ranged from 4.5 to 34.2 psu, the N-Total ranged from 0.01% to 0.12%, P ranged from 36.68 to 266.54 c mol/kg, and K ranged from 195.99 to 1618.05 c mol/kg (relevés and sampling were performed in the dry season). More than 60% of soils sampled (13 of 22 sites) were dominated by sand and a combination of a sandy-loam texture (Table 5). Clearly, loamy/clayey sites were few: examples are Batu Beriga, Kota Kapur, Sampur Beach, Tanjung Punai, and Tanjung Pura. These sites were characterised by the highest clay percentage.

Canonical Correspondence Analysis (CCA) was used to examine the relationship between mangrove species and environmental factors (Figure 4). The CCA results indicated that the abundance of true mangrove species was significantly associated ($p < 0.05$) with measured edaphic factors. However, these factors only explained a limited proportion of the variability in species composition. Specifically, Axis 1 and 2 accounted for just 37.2%. The species in the first quadrant (Figure 4, upper left) were primarily associated with

higher levels of potassium (K), lower salinity, lower sand proportion in the substrate, and soils characterised by a higher percentage of clay and silt. In this quadrant, *Sonneratia alba* abundance was associated mostly with a higher clay percentage and potassium (K). Meanwhile, *Aegiceras corniculatum*, *Bruguiera cylindrica*, *Bruguiera parviflora*, and *Excoecaria agallocha* abundance was linked to a higher silt proportion and reduced saline soil. It is important to note that edaphic factors may be responsive to other causes, such as location, elevation, and geomorphology, which independently determine mangrove presence and abundance. Thus, the CCA results indicate that only a few species displayed a clear association with more than one edaphic factor.

**Table 5.** Edaphic factors at each sampling site on Bangka Island.

| Location | Sand (%) | Silt (%) | Clay (%) | Texture Class | Salinity (psu) | N-Total (%) | P (c mol/kg) | K (c mol/kg) |
|---|---|---|---|---|---|---|---|---|
| Batu Beriga | 31.41 | 49.24 | 19.25 | Loam | 30 | 0.05 | 110.73 | 737.27 |
| Dante Island | 60.59 | 27.12 | 12.29 | Sandy Loam | 31 | 0.06 | 40.25 | 497.05 |
| Ibul Island | 39.6 | 40.58 | 19.82 | Loam | 30 | 0.07 | 180.07 | 630.81 |
| Jebu Laut | 60.1 | 30.98 | 8.92 | Sandy Loam | 30 | 0.07 | 76.12 | 365.03 |
| Kepoh | 3.27 | 55.13 | 41.6 | Silty Clay | 34.2 | 0.03 | 131.13 | 1132.27 |
| Kota Kapur | 2.85 | 29.31 | 67.84 | Clay | 30 | 0.01 | 169.59 | 1618.05 |
| Lubuk Besar | 92.3 | 5.57 | 2.13 | Sand | 30 | 0.07 | 48.63 | 195.99 |
| Merbau Beach | 38.7 | 24.03 | 37.27 | Clay Loam | 24.5 | 0.06 | 72.6 | 754.86 |
| Pangkul Beach | 72.2 | 11.34 | 16.46 | Sandy Loam | 4.5 | 0.04 | 56.76 | 263.61 |
| Panjang Island | 90.37 | 6.48 | 3.15 | Sand | 31 | 0.04 | 154.79 | 439.52 |
| Pejem Beach | 91.17 | 7.71 | 1.12 | Sand | 29 | 0.04 | 63.43 | 356.38 |
| Sadai | 71.33 | 18.46 | 10.21 | Sandy Loam | 30 | 0.12 | 49.63 | 426.12 |
| Sampur Beach | 9.19 | 32.35 | 58.46 | Clay | 15 | 0.03 | 88.24 | 529.22 |
| Seniur Island | 90.42 | 7.19 | 2.39 | Sand | 32.6 | 0.03 | 266.54 | 295.23 |
| Sukal | 84.82 | 2.59 | 12.59 | Loamy Sand | 30 | 0.05 | 57.78 | 351.85 |
| Tanjung Bajun | 96.65 | 2.39 | 0.96 | Sand | 30 | 0.04 | 71.26 | 388.23 |
| Tanjung Labu | 96.23 | 2.62 | 1.14 | Sand | 30.8 | 0.01 | 148.51 | 264.59 |
| Tanjung Pao | 82.764 | 3.05 | 14.19 | Sandy Loam | 23.5 | 0.03 | 73.36 | 444.8 |
| Tanjung Punai | 2.39 | 42.45 | 55.16 | Silty Clay | 31 | 0.06 | 96.13 | 1055.81 |
| Tanjung Pura | 4.7 | 48.26 | 47.04 | Silty Clay | 29 | 0.04 | 104.5 | 1066.08 |
| Tanjung Sunur | 32.31 | 64.96 | 2.73 | Silt Loam | 30 | 0.07 | 45.96 | 386.96 |
| Tanjung Tada | 74.62 | 8.08 | 17.3 | Sandy Loam | 24.6 | 0.07 | 36.68 | 479.25 |

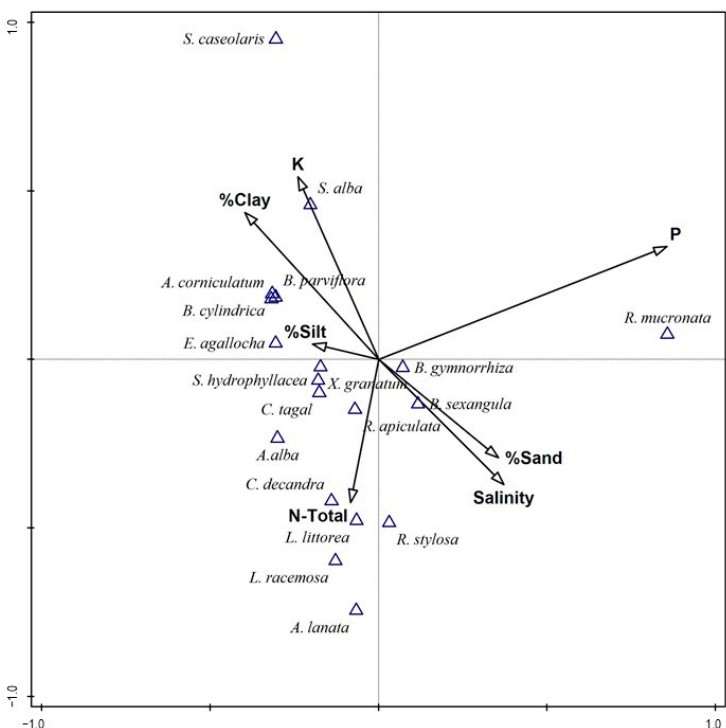

**Figure 4.** Canonical Correspondence Analysis (CCA) biplots of mangrove species (triangles) and environmental variables (arrows) for all sampled sites and true mangrove species abundance of Bangka island.

## 4. Discussion

Our study aimed to address several key questions relating to mangroves on Bangka Island. First, we investigated species richness, presence/absence, diversity, and species composition throughout representative sites along Bangka Island's coastline. Secondly, we analysed the pattern of the mangrove zone along the transects perpendicular to the water line. We then explored potential variations in composition and diversity between the sampling sites. Lastly, we examined the influence of various edaphic factors, including soil texture, pore-water salinity, N-total, P, and K, on mangrove composition. Our findings revealed that the mangroves on Bangka Island could be categorised into distinct groups, exhibiting significant differences in composition and species abundance between the sites of the resp. groups. We found that a small number (37.2%) of measured edaphic factors were significantly associated with the abundance of true mangrove species on Bangka Island (Figure 4). Through satellite imagery, we observed a probable zonation in the widest forest, in which a 100 m monospecific transect did not cover the full variation of the ecotone in the field. Conversely, in sites with mixed species, the zonation pattern was less evident. This indicates that other factors beyond the studied edaphic factors may contribute to the composition, structure and zonation of Bangka Island mangroves.

### 4.1. Mangrove Species Composition and Distribution on Bangka Island

The recorded species in our study represent most common species present within the longitudinal range of 90° E to 120° E [56], which comprises the coordinates of Bangka Island. A total of 21 species of true mangrove and two associates (*sensu* Kitamura et al. 1997) [54] were identified from various sampling sites on Bangka Island (Table 1). By considering the findings from previous studies on Bangka Island (Table 2), these data include approximately 70% of the species known to exist in Indonesia [60], and hence the island is representative at that level for the area with the second highest mangrove species richness worldwide (26–35), as compiled by Polidoro et al. (2010) [2]. Notably, our findings indicate that there are no specific species (assemblages) restricted to either the southern or northern, the western or eastern parts of Bangka Island.

In the Bangka Barat Regency, nine species were reported [26]. However, our study identified 12 species and half of them had not been reported previously. These unreported species include *Avicennia alba*, *Bruguiera sexangula*, *Ceriops tagal*, *Excoecaria agallocha*, *Lumnitzera littorea*, and *Pandanus tectorius*. Moreover, our findings revealed that other species such as *Acanthus ilicifolius*, *Acrostichum speciosum*, *Aegiceras corniculatum*, *Bruguiera cylindrica*, *Bruguiera parviflora*, *Bruguiera sexangula*, *Ceriops decandra*, *Nypa fruticans*, and *Scyphiphora hydrophyllacea* had also not been encountered or reported in previous studies in the Bangka Selatan Regency [45,61–63]. Likewise, studies in the Bangka Tengah Regency [24,30,31,42,43,64] had not reported species such as *Aegiceras corniculatum*, *Bruguiera cylindrica*, *Ceriops decandra*, and *Scyphiphora hydrophyllacea*. Furthermore, in the Bangka Regency, this study revealed the presence of *Acrostichum speciosum*, *Aegiceras corniculatum*, *Bruguiera parviflora*, *Excoecaria agallocha*, *Lumnitzera littorea*, *Pandanus tectorius*, and *Xylocarpus granatum*, which have not been encountered or reported before [27,44]. In addition to the mangrove presence reported in the previous studies on Bangka Island, *Lumnitzera littorea* and *Lumnitzera racemosa* have never been found in the same habitat and location due to non-established ecological character differences [55]. But, we recorded both species in one of our sampling sites on Bangka Tengah (Lubuk Besar). Additionally, we recorded *Avicennia marina*, *Acrostichum aureum*, *Heritiera littoralis*, and *Pemphis acidula* outside the observation plots (Table 2).

The differences in the species recorded between this study and previous studies are likely due to variations in the sampling locations within the same regency, sampling year and the methods employed. Nonetheless, this highlights several key points. Firstly, our research findings mostly corroborate previous studies, indicating that the composition of mangrove species varies significantly across nearly all sampling locations on Bangka Island. Secondly, the discrepancies in the method used to determine stem diameter classes

and plot size could potentially result in the omission of certain species from the earlier records. Thirdly, the different sampling years might also explain some discrepancies in the species records from Bangka Island, and certain presences due to plantation. Lastly, some of the species are rare and hence may be missed in specific campaigns in specific areas. In conclusion, Bangka Island has a wide variety of species and it is possible that not all species have been documented in all studies.

We identified two species that are included in the IUCN global Red List of Threatened Species. *Avicennia lanata* was identified as Vulnerable (VU) and *Ceriops decandra* as Near Threatened (NT) among all identified mangrove species. In addition, *Bruguiera hainesii* [30] and *Sonneratia ovata* [25], recorded in Kurau Timur Village (Bangka Tengah) and Pangkalpinang, respectively, were not observed during the survey but are classified as Critically Endangered (CR) and Near Threatened (NT), respectively. Therefore, without effective conservation efforts aiming to mitigate human impacts on their habitats, species at high risk of extinction face the possibility of disappearing, and the presence of rare species may go unnoticed. The loss of individual species may affect biodiversity, ecosystem function, and human livelihoods, especially in areas with low diversity and high habitat loss [2].

The species composition within the sampling sites can be categorised into five groups based on 40% similarity (Figure 2). This indicates that the mangrove composition was different in almost all sampling sites, rendered possible by the high species richness of the biogeographical region, but also reflecting various coastal settings. Group (I) comprises eight sites: Dante Island, Ibul Island, Jebu Laut, Kepoh, Merbau Beach, Sadai, Tanjung Pao, and Tanjung Tada. The sampling site typology in this group varied between the estuary, open coast, and lagoon [8]. Within this group, *Rhizophora apiculata*, *Ceriops tagal*, and *Xylocarpus granatum* contributed to the overall grouping, with *Rhizophora apiculata* being the major contributor (38%), meaning that these species have the highest abundance in this group. Among these locations, Dante Island, Tanjung Tada, Jebu Laut, and Sadai have a sandy loam substrate, where *Rhizophora apiculata* and *Ceriops tagal* exhibit a high abundance compared to other species at each site. The soil texture might be suitable for the growth of *Rhizophora apiculata* and *Ceriops tagal*. *Sonneratia alba* and *Avicennia alba* are also present in this substrate type, albeit in a lower abundance. Additionally, *Rhizophora apiculata* shows the highest abundance in three other sites (Ibul Island, Kepoh, and Merbau Beach) with loam, silty clay, and clay loam substrates, respectively. The species in this group are commonly found in the middle intertidal zone at higher elevations. The elevation is linked to the inundation period, where the elevated ground may only be submerged infrequently and still get freshwater input from the hinterland or rainfall. A similar formation of the Rhizophora area was also recorded in Telok Sekanak (Bangka) [39], where Rhizophora species appeared to grow at higher ground and in less flat areas.

Group (II) consists of Batu Beriga, Pejem Beach, Sampur Beach, Tanjung Bajun, Tanjung Labu, Tanjung Pura, and Tanjung Sunur sites. Most of these sites share two common species, *Rhizophora apiculata*, and *Sonneratia alba*, except for Pejem Beach, which only has *Rhizophora apiculata* in common with the other sites. The substrate within this group varies, including silt loam, clay, sand, silty clay, and loam. This indicates that *Rhizophora apiculata* is the most abundant species in almost all sites, regardless of substrate type. Group (III) comprises the Panjang and Seniur Island locations with a sandy substrate. The high abundance of *Rhizophora mucronata* contributes to the grouping within this group. *Rhizophora apiculata* and *Bruguiera gymnorrhiza* are also present but in a lower abundance. The higher sand fraction in the substrate might be more suitable for the growth of *Rhizophora mucronata* than *Rhizophora apiculata* and *Bruguiera gymnorrhiza*. This finding aligns with a recent study in the city of Bengkulu [81], which verifies that *Rhizophora mucronata* is more adaptable to hard and sandy soils than *Rhizophora apiculata*.

Group (IV) includes Kota Kapur, Pangkul Beach, Sukal, and Tanjung Punai. These locations have substrate textures of silty clay, sandy loam, clay, and loamy sand, respectively. Among these sites, *Sonneratia alba* is the most abundant species. These areas likely have

lower elevations, leading to periodic inundation. This finding is supported by a previous study [39], in which the presence of *Sonneratia alba* extending seaward indicates regular inundation by saltwater and progressive silting. In addition, Bangka Island experiences diurnal tides [51], resulting in daily inundation. The combined environmental condition of daily inundation due to diurnal tides and the appropriate elevation creates a favorable growth environment for *Sonneratia alba*, which prefers high salinity and muddy soil. Furthermore, Pangkul Beach, with a sandy loam substrate consisting of approximately 72.22% sand, has the highest abundance of *Sonneratia alba*. This finding contradicts Group (I), where *Sonneratia alba* was only recorded in small abundance in sandy loam substrates. The substrate type in Pangkul Beach is believed to be influenced by nearby active tin mining operations and tin tailings. Since the substrate is one of the important factors in shaping mangrove composition, it is possible that in the future, there will be changes in the mangrove composition of this area.

Group (V) solely consists of one site, Lubuk Besar. It stands as an outlier compared to the other four groups, indicating that its mangrove composition is significantly different. This site is dominated by *Lumnitzera racemosa*. The mangrove typology in this location is lagoon carbonate [8] and soil substrate dominated by sand. The formation of mangroves in Lubuk Besar started behind beach vegetation. Within this site, the Rhizophoraceae family is recorded in low abundance. The dominance of *Lumnitzera racemosa* in this group, along with the presence of associate species outside the observation plots, such as *Terminalia catappa* and *Casuarina equisetifolia*, gives evidence of elevated ground. In addition to elevation, the climate of Bangka Island provides suitable growth conditions for these species: under a high rainfall rate, the elevated area (above the diurnal tide level) washes out frequently, resulting in a lower substrate salinity. Furthermore, the absence of observed pioneer species such as *Sonneratia alba* aligns with the characteristics of the topography. This species prefers to grow in areas that are regularly inundated by saltwater. Moreover, the mangrove forest in this site appears to be clumped. This clumped distribution is somewhat related to the (post) tin mining activities as this site is in close proximity to an abandoned tin mining pond. The extensive changes in soils and vegetation disturbance caused by the intense tin mining operations likely contribute to the observed effects [34,36].

*4.2. Zonation Pattern along the Observation Transect*

A segmentation analysis of the mangrove areas on Bangka Island (Figure 3) suggests the presence of a zonation pattern in apparently monospecific sites like Kota Kapur. However, this zonation pattern was not detected during field observations due to the limited length of the transect, which was only 100 m: this transect length was insufficient to capture the mangrove zonation in that area, considering that the width of the mangrove area is approximately 1.5 km [9]. On the other hand, for sites with mixed species along the observation transects, the variation in environmental conditions contributes to a poorly delineated zonation. There is no consistent geographical pattern in which specific species or mangrove compositions are found in different sections of the coastline. We can outline elevational or positional presences of species, such as *Ceriops tagal* and *Xylocarpus granatum*, which confer a structure or weakly recurring zonation. Nevertheless, none of the elevational presences are so strict that there is a general and clear zonation throughout Bangka (tidal amplitude between 2 m and 3 m) [51]. The species richness also shows that distinct species can occupy a certain elevation, giving a rich mosaic structure, with major differences amongst sites.

The zonation pattern may differ between biogeographical areas, and also within the Indo-Pacific region. For example, species zonation in Sri Lanka is seldom observed [70], while on the Andaman islands, there is no evidence of differences in the zonation pattern [79]. However, on Kerala [80] and Belitung [77], distinct species zoning patterns have been reported at each specific location within the region. Furthermore, the earliest attempt to monitor mangroves on Bangka Island in 1930 [39] revealed that vegetation in floodplain forests changes according to growth site conditions. Different areas provide favourable conditions for the growth of specific tree species. As a result, factors such as elevated

ground, site-specific edaphic conditions, and anthropogenic activities such as tin mining operations might influence the zonation of mangrove vegetation (Figure 5). These findings highlight that the unique character of each site has an impact on the zonation of mangrove species on Bangka Island, but this must be further corroborated or tested.

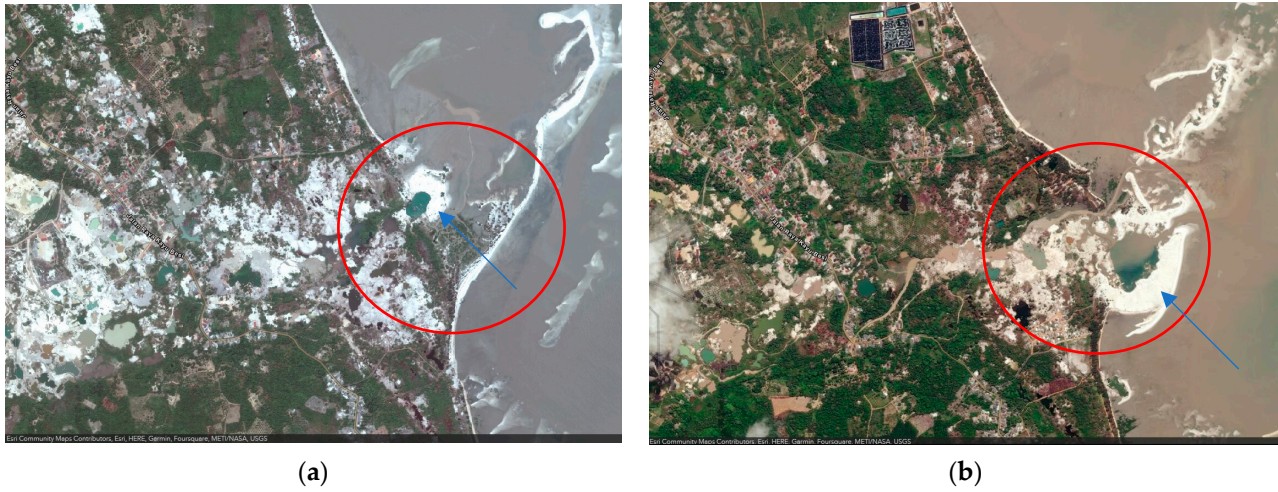

(**a**)　　　　　　　　　　　　　　　　　　　　　　　　　(**b**)

**Figure 5.** Comparative visual impact of tin mining on the potential alteration of mangrove zonation at Pangkul Beach; the red circle highlights the coastal area's change, the blue arrow indicates the tin mine tailings: (**a**) September 2011; (**b**) July 2019.

*4.3. Diversity of True Mangrove Species*

Our findings indicate that the mangrove forest on Bangka Island exhibits a low diversity pattern (Shannon index < 2) against a background of high species richness, characteristic of mangroves and the biogeographical region, and despite the effective presence of various recorded species. Similar low Shannon index values have been observed in other regions such as Pangkalpinang (Bangka Island, Indonesia) [25], the Gulf of Khambat (India) [78], Sibuti (Malaysia) [74], Air Telang Protected Forest (South Sumatra, Indonesia) [75] and Vidattaltivu (Sri Lanka) [69], each with varying numbers of identified species (Table 4). In contrast, the mangrove forest on Belitung Island, the neighbour island of Bangka, exhibits a higher Shannon index ranging from 2.19 to 2.7 [77], with a similar species composition and richness recorded. The Shannon index value can be attributed to several potential factors. First, the species composition within the community and the distribution proportions of each individual species can influence the Shannon index value [75]. Second, the maturity of the forest plays a role, as mature forests tend to have a greater canopy cover, limiting light penetration and thereby impeding the colonisation of other species [66]. Third, a low Shannon index indicates the degraded nature of the mangrove forest [82]. Lastly, site characteristics may contribute to differences in species richness and diversity.

*4.4. The Relation between Edaphic Factors and Mangrove Abundance*

The five mangrove groups identified using the Bray–Curtis similarity cluster analysis are clearly differentiated by their species composition and environmental characteristics. However, when we analysed the measured edaphic factors in the Canonical Correspondence Analysis (CCA), we discovered that they could only account for 37.2% of the variation in species composition. This finding contradicts previous studies that clearly demonstrated the influence of edaphic factors such as soil texture, nutrients, soil pH, temperature, and salinity [77,79,83,84] on mangrove composition. But, there are plausible explanations for the observations recorded in our study. First, the presence of mangrove associates at the sampling sites indicates that factors beyond the measured edaphic variables account for 62.8% of the unexplained variation in mangrove composition on Bangka Island. Second,

many true mangrove species exhibit a wide ecological tolerance with regard to edaphic factors, and are not strictly limited to very narrow environmental ranges [83].

In addition to the measured edaphic factors, there are various other factors that influence the distribution and composition of mangroves, accounting for the remaining 62.8% of unexplained variations, factors that are not measured in this study. Examples of these factors include the redox potential Eh [79], geomorphological features, propagule release and dispersal [85], the bioclimate, sea surface temperature [86], sea level variation [87], ocean currents, tidal inundation, and wave action [88]. The interaction of local natural and anthropogenic factors [68], as well as the site's history, also play a role in shaping the distribution and composition of mangrove vegetation.

Furthermore, the pattern of ocean currents in the waters surrounding Bangka Island potentially contributes to the variations in mangrove composition and distribution. These currents exhibit a consistent north-to-south flow during the West Monsoon (December to February), and the opposite direction during the East Monsoon (June to August) [89]. The coastal areas of Bangka Island exhibit current velocities ranging from 0 to 0.6 m/s [90], which are slightly higher compared to the Bangka Strait, whose velocities range from 0 to 0.452 m/s [91]. Notably, research from [92] demonstrates that these currents can disperse particles within a radius of 16 miles. Given this understanding, it is reasonable to infer that mangrove propagules can be carried by ocean currents to various parts of Bangka Island over a distance similar to these particles; yet, there is not a homogeneous composition along Bangka's coastline, nor does the range of every species cover the entire coast (where mangroves are present).

Aside from the ocean current, sedimentation also plays a significant role in the distribution and composition of the mangrove species. Sedimentation processes are prevalent in the majority of river estuaries on Bangka Island. A recent study on Bangka Tengah recorded a substantial sedimentation rate of 0.58 cm/day in the Kurau River estuary [93], which promotes land expansion and the growth of new mangrove ecosystems in this estuary. For instance, the upstream sediment influx, primarily caused by mining operations performed by local small-scale miners, which is known as unconventional tin mining (sensu Nurtjahya and Agustina, 2015) [34], can cause increased sediment accumulation downstream, resulting in higher levels of siltation in the water (Figure 6). This process also provides favourable conditions for land expansion, creating opportunities for mangrove propagules to establish themselves and grow on the newly available substrate. Additionally, these sedimentation processes may contribute to the proliferation of mangroves through the dispersal of propagules from existing populations along riverbanks. On the other hand, the deposition of sediment from tin tailings can adversely affect existing mangrove stands, potentially leading to dieback [27] and a change in species composition. The sediment can clog the pneumatophores and disrupt the respiration process of mangroves, but the chemical nature of its origin and its impacts remain to be studied, particularly in tin-mine-affected areas.

*4.5. Study Limitation*

A potential limitation of this study is that, in view of the high species richness of the wider region and the rarity of some species, a higher sample number would have improved the coverage and representativity. The transect length used for data collection may not have been sufficient to fully represent the species diversity and ecological patterns across all coastal ecotones of Bangka Island. Hence, the observed differences in the recorded species over time, as well as the apparent zonation in some very wide forests not adequately covered in the field campaign, point to the need for complementary campaigns. The variety of coastal settings on Bangka island, mirrored in the great diversity of mangrove forest types, calls for a typology describing the coastal types; this may be a higher priority than adding new edaphic factors, which are very labour-intensive during sampling and analysis. In a pioneering work based on high-quality aerial photography and keen ground observation, Kint [39] offers a number of narrative explanations for different mangrove types (in the north of Bangka Island), which are still relevant as a guideline for such

research today. In addition, to apply the findings of this study for rehabilitation purposes, conducting further research and analysis involving additional parameters is required.

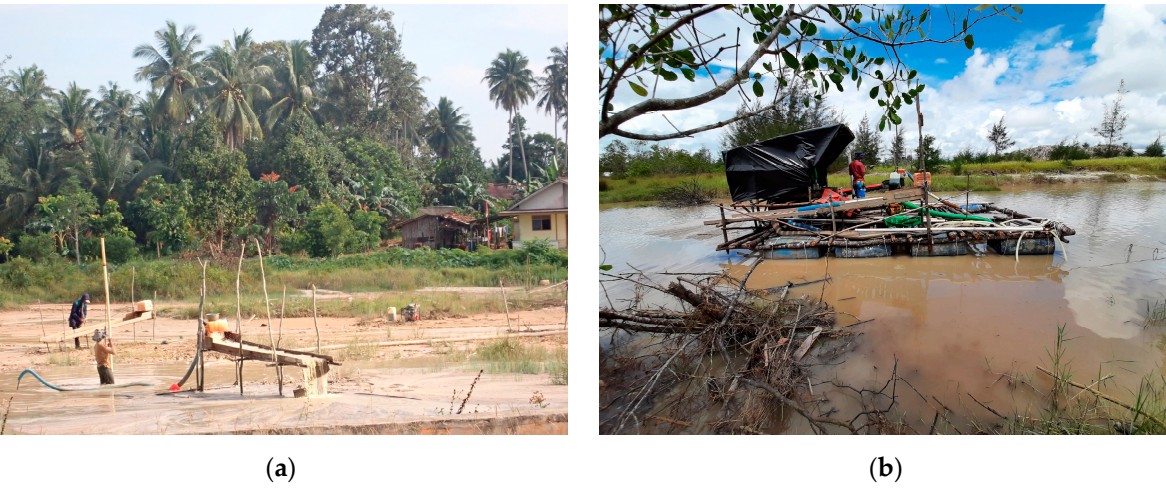

(**a**)                                                                                    (**b**)

**Figure 6.** Examples of so-called 'unconventional tin mining' on Bangka Tengah: (**a**) Tin mining op-eration in an inland area (August 2022); (**b**) Tin mining operation in the adjacent offshore area, which is located near the mangrove area (September 2021).

## 5. Conclusions

Amongst the 22 sites investigated throughout coastal Bangka, we have discerned their diversity and differences, without an island-wide geographical pattern and without a strongly determining ecological setting, at least for the parameters assessed. This suggests that the similarities and differences among the sites are spread evenly along the coast. In addition, the overall richness and diversity are remarkably high, encompassing approximately 70% of the most commonly recorded species in Indonesia. None of the species recorded in this study were unexpected. Besides the threat of the impact of illegal tin mining, which has increased over the last two decades, mangrove loss through other practices may wipe out or floristically change the local vegetation (including a few Red-listed species), and there may not be a similar type elsewhere on the island. Against the background of diversity, there is little intersite redundancy, and effective conservation can still preserve Bangka's wealth.

This calls for the cautious establishment of forest management practices for sites along the entire coast. The unique and distinct nature of each site on Bangka makes the task of protecting and restoring the area more complex and challenging. Simply protecting a few sites will not be enough to represent the mangrove diversity of Bangka Island. Instead, carefully selecting protected sites is necessary to ensure the preservation of the complete range of species and ecological features found on Bangka, as well as the processes forming and changing them. By studying the occurrence of specific species, we can identify the biotope and gain insights into the prevailing vegetation and topographical characteristics associated with growth sites. Furthermore, the findings from this research, combined with earlier studies, hold significant value for the on-site conservation and management of mangroves on Bangka Island and the mangrove hotspot Indonesia. However, for future research addressing the mangrove distribution on Bangka, it is essential to include a larger number of sites that cover the complete species richness. Additionally, the coastal environmental settings of the island, such as (first) the coast type, edaphic factors, typology, elevation, and tidal inundation, need to be considered, along with the presence of tin resources and their exploitation, with its potential negative impacts.

**Author Contributions:** Conceptualization, S.P.S., N.K. and F.V.C.; methodology, S.P.S., N.K. and F.V.C.; formal analysis, S.P.S.; investigation, S.P.S.; resources, A.P. and M.R.M.; data curation, S.P.S., A.P. and M.R.M.; writing—original draft preparation, S.P.S.; writing—review and editing, S.P.S., N.K., A.P., M.R.M. and F.V.C.; visualization, S.P.S.; supervision, N.K. and F.V.C. All authors have read and agreed to the published version of the manuscript.

**Funding:** This research was funded by the operational research fund from the Ministry of Education, Culture, Research, and Technology of the Republic of Indonesia, through Beasiswa Program Pascasarjana Luar Negeri (BPPLN).

**Data Availability Statement:** The data provided in this research can be obtained by contacting the corresponding author.

**Acknowledgments:** The first author would like to thank their colleagues and students from the Department of Marine Science of Universitas Bangka Belitung (Adelia, Alif, Ardad, Husin, Ikhwan, Linda, Mia, Ramadhan, Rara, Syahrin, and Wahyu) for their assistance during fieldwork. We thank the Marine Science and Aquatic Resources Management Laboratory of Universitas Bangka Belitung for facilitating the field data collection. Special thanks go to Tengku Zia Ulqodry, Mengxi Wang, Nur Annis Hidayati, Susana Endah Ratnawati, Gladys and Anik Juli for their feedback on the manuscript and submission process.

**Conflicts of Interest:** The authors declare no conflict of interest.

## Appendix A

**Table A1.** Mangrove sampling sites on Bangka Island.

| ID | Location | Regency | Typology | Longitude (E) | Latitude (S) | Mangrove Thickness (m) | Site Condition |
|---|---|---|---|---|---|---|---|
| BT1 | Batu Beriga | Bangka Tengah | open coast carbonate | 106°45′52.265″ | 2°35′54.852″ | 125.18 | Shrimp pond nearby |
| BA1 | Dante Island | Bangka | estuarine terrigenous | 105°46′11.172″ | 1°44′17.963″ | 228.59 | No anthropogenic activities detected |
| BS1 | Ibul Island | Bangka Selatan | open coast carbonate | 106°46′16.518″ | 2°53′37.972″ | 444.49 | No anthropogenic activities detected |
| BB1 | Jebu Laut | Bangka Barat | estuarine terrigenous | 105°30′55.811″ | 1°34′1.326″ | 283.57 | Settlement area nearby |
| BS2 | Kepoh | Bangka Selatan | open coast carbonate | 106°35′27.776″ | 2°55′32.866″ | 426.18 | No anthropogenic activities detected |
| BA2 | Kota Kapur | Bangka | open coast terrigenous | 105°47′25.156″ | 2°13′45.966″ | 1613.81 | No anthropogenic activities detected |
| BT2 | Lubuk Besar | Bangka Tengah | lagoonal carbonate | 106°39′25.006″ | 2°32′14.154″ | 100.00 | Fishing area, mining |
| BS3 | Merbau Beach | Bangka Selatan | open coast carbonate | 106°26′21.106″ | 2°59′25.447″ | 170.00 | Shrimp pond, tourist and settlement area nearby |
| BT3 | Pangkul Beach | Bangka Tengah | open coast carbonate | 106°12′44.111″ | 2°13′42.982″ | 150.00 | Tourism and mining area nearby |
| BT4 | Panjang Island | Bangka Tengah | open coast carbonate | 106°16′20.244″ | 2°9′32.378″ | 277.19 | No anthropogenic activities detected |
| BA3 | Pejem Beach | Bangka | open coast carbonate | 105°54′55.433″ | 1°31′21.806″ | 100.00 | Tourism and settlement area nearby |
| BS4 | Sadai | Bangka Selatan | open coast carbonate | 106°43′21.720″ | 3°0′19.620″ | 398.29 | Industrial area nearby |
| BT5 | Sampur Beach | Bangka Tengah | open coast carbonate | 106°10′45.847″ | 2°9′5.782″ | 190.00 | Tourism and mining area nearby |
| BS5 | Seniur Island | Bangka Selatan | open coast carbonate | 106°47′11.508″ | 2°50′3.948″ | 710.72 | No anthropogenic activities detected |
| BB2 | Sukal | Bangka Barat | open coast terrigenous | 105°20′52.796″ | 2°7′26.108″ | 270.00 | Settlement area and port nearby |
| BB3 | Tanjung Bajun | Bangka Barat | open coast carbonate | 105°36′7.488″ | 1°33′0.155″ | 219.98 | No anthropogenic activities detected |
| BS6 | Tanjung Labu | Bangka Selatan | open coast carbonate | 106°49′56.118″ | 2°53′58.708″ | 336.59 | No anthropogenic activities detected |
| BS7 | Tanjung Pao | Bangka Selatan | open coast carbonate | 106°37′24.121″ | 2°57′40.381″ | 150.00 | Shrimp pond, tourism and settlement area nearby |
| BB4 | Tanjung Punai | Bangka Barat | open coast terrigenous | 105°18′3.892″ | 2°9′15.761″ | 3585.60 | No anthropogenic activities detected |
| BT6 | Tanjung Pura | Bangka Tengah | open coast terrigenous | 105°52′33.258″ | 2°25′43.403″ | 2027.91 | No anthropogenic activities detected |
| BA4 | Tanjung Sunur | Bangka | estuarine terrigenous | 105°45′56.225″ | 1°46′10.168″ | 300.83 | No anthropogenic activities detected |
| BB5 | Tanjung Tada | Bangka Barat | open coast terrigenous | 105°26′47.425″ | 2°7′58.840″ | 692.86 | No anthropogenic activities detected |

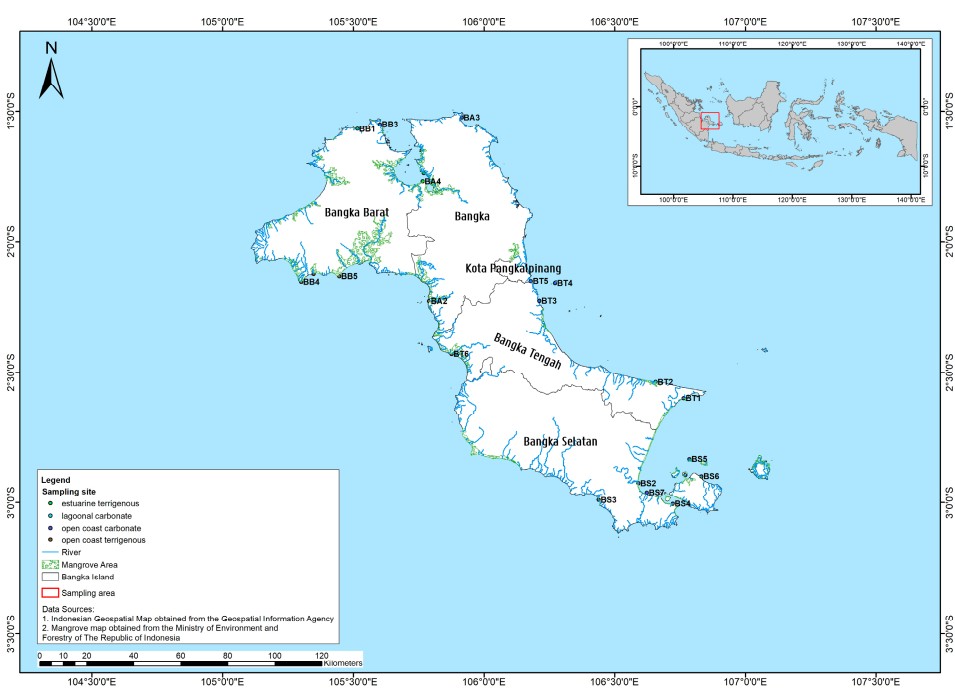

**Figure A1.** Study area map showing the observation location of the mangroves on Bangka Island.

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
