# Peer review of "Unveiling the Diversity of Bangka Island’s Mangroves: A Baseline for Effective Conservation and Restoration"

_forests, doi:10.3390/f14081666_

Round 1
Reviewer 1 Report
General comments
I found the manuscript interesting as it provides good baseline data, the current status of mangrove species richness, and edaphic factors determining their distribution. The study provides insights into what needs to be done to conserve mangrove forests, especially given their value to the ecosystem and the people.
Specific comments
Line 39: What is the unit of 2100 value?
Lines 48 and 51: 3.364.080 and 182,091. Be consistent, and I suggest using commas instead of full stops.
The scientific names of the last two species in Table 1 are not in italics.
Table 2: I suggest that you replace ‘’Species number” with “Number of individuals”.
Statistical analyses: Can the authors explain why they did not do statistical analyses for parameters such as tree density, diversity, salinity, N-Total, P and K to determine significant differences across sites?
Line 350: Move this to the results section, including Table 4.
Figure 5 should be removed from the discussion to the results section. Alternatively, the authors can combine results and discussion to avoid presenting results in the discussion section.
Table 5: same comments as the above.
Lines 550-551: It is better to say December to February and June to August.
The language is good.
Reviewer 2 Report
This is a good and interesting MS on these issues, but challenging to apply in rehabilitating mangroves with respect to these results.
I suggest using several parameters that can be used by mangrove rehabilitation implementers that do not require laboratory analysis.
Additional information is needed to improve this ms, such as mangrove thickness in each sampling location, research area (Table 5), notation at each sampling location (Tabel A1 and Figure A1).

The English are good.
